# Cost–Utility of Mindfulness-Based Stress Reduction for Fibromyalgia versus a Multicomponent Intervention and Usual Care: A 12-Month Randomized Controlled Trial (EUDAIMON Study)

**DOI:** 10.3390/jcm8071068

**Published:** 2019-07-20

**Authors:** Adrián Pérez-Aranda, Francesco D’Amico, Albert Feliu-Soler, Lance M. McCracken, María T. Peñarrubia-María, Laura Andrés-Rodríguez, Natalia Angarita-Osorio, Martin Knapp, Javier García-Campayo, Juan V. Luciano

**Affiliations:** 1Group of Psychological Research in Fibromyalgia & Chronic Pain (AGORA), Institut de Recerca Sant Joan de Déu, 08950 Esplugues de Llobregat, Spain; 2Teaching, Research & Innovation Unit, Parc Sanitari Sant Joan de Déu, 08830 Sant Boi de Llobregat, Spain; 3Primary Care Prevention and Health Promotion Research Network, RedIAPP, 28029 Madrid, Spain; 4Department of Clinical Psychology and Psychobiology (Section Personality, Assessment and Psychological Treatments), University of Barcelona, 08193 Barcelona, Spain; 5The London School of Economics and Political Science (LSE), London WC2A 2AE, UK; 6Department of Psychology, Uppsala University, SE-751 05 Uppsala, Sweden; 7Primary Health Centre Bartomeu Fabrés Anglada, SAP Delta Llobregat, Unitat Docent Costa de Ponent, Institut Català de la Salut, 08850 Gavà, Spain; 8Centre for Biomedical Research in Epidemiology and Public Health, CIBERESP, 28029 Madrid, Spain; 9Fundació IDIAP Jordi Gol I Gurina, 08007 Barcelona, Spain; 10Department of Psychiatry, Miguel Servet Hospital, Aragon Institute of Health Sciences (I+CS), 50009 Zaragoza, Spain

**Keywords:** fibromyalgia, cost–utility, cost-effectiveness, quality-adjusted life years

## Abstract

Fibromyalgia (FM) is a prevalent, chronic, disabling, pain syndrome that implies high healthcare costs. Economic evaluations of potentially effective treatments for FM are needed. The aim of this study was to analyze the cost–utility of Mindfulness-Based Stress Reduction (MBSR) as an add-on to treatment-as-usual (TAU) for patients with FM compared to an adjuvant multicomponent intervention (“FibroQoL”) and to TAU. We performed an economic evaluation alongside a 12 month, randomized, controlled trial; data from 204 (68 per study arm) of the 225 patients (90.1%) were included in the cost–utility analyses, which were conducted both under the government and the public healthcare system perspectives. The main outcome measures were the EuroQol (EQ-5D-5L) for assessing Quality-Adjusted Life Years (QALYs) and improvements in health-related quality of life, and the Client Service Receipt Inventory (CSRI) for estimating direct and indirect costs. Incremental cost-effectiveness ratios (ICERs) were also calculated. Two sensitivity analyses (intention-to-treat, ITT, and per protocol, PPA) were conducted. The results indicated that MBSR achieved a significant reduction in costs compared to the other study arms (*p* < 0.05 in the completers sample), especially in terms of indirect costs and primary healthcare services. It also produced a significant incremental effect compared to TAU in the ITT sample (ΔQALYs = 0.053, *p* < 0.05, where QALYs represents quality-adjusted life years). Overall, our findings support the efficiency of MBSR over FibroQoL and TAU specifically within a Spanish public healthcare context.

## 1. Introduction

Fibromyalgia (FM) is a disabling syndrome of unknown etiology mainly characterized by chronic widespread musculoskeletal pain, fatigue, stiffness, sleep problems, perceived cognitive dysfunction, and mood disturbances [1]. It is usually diagnosed in women aged between 30 and 50 years old and has an estimated prevalence of around 2% in the general population [2,3]. Health-related quality of life is significantly lower for people with FM compared to the general population and similar or lower than those seen for other medical conditions such as osteoarthritis, rheumatoid arthritis, or osteoporosis [4].

FM is a costly syndrome for both healthcare funders and society in general. It is the chronic pain condition with the highest rates of unemployment, sick-leave, claims for incapacity benefits, work absenteeism, and per-patient costs [5]. FM patients’ direct costs have been described as three times higher than those for patients with other pathologies but similar sociodemographic characteristics [6]. Regarding indirect costs, the range of women with FM who are able to preserve their jobs has been reported to be between 34% and 77% [7,8], and reducing work hours due to the impact of symptoms is a common practice among patients with FM [9]. Altogether, FM is second only to irritable bowel syndrome in its contribution to the approximately $300B in costs that inflammatory diseases and related chronic syndromes are expected to generate in the US in coming years [10]. There is a need, therefore, to optimize the development and the implementation of treatments for FM that, next to early accurate diagnoses and methods to support treatment-adherence, would help to address this burden [10].

To date, no cure has been found for FM, although different pharmacological (pregabalin and noradrenaline reuptake inhibitors) and non-pharmacological interventions (aerobic exercise, cognitive-behavioral therapy (CBT), multicomponent therapy, and “third-wave” CBT such as mindfulness-based interventions (MBIs) or Acceptance and Commitment Therapy (ACT)) have demonstrated some benefits for reducing the impact of the symptoms and increasing quality of life [11,12]. The pharmacological approach is most common for FM despite its limited effectiveness. In fact, non-pharmacological treatments appear to show effects in more separate symptom domains as compared with pharmacological treatments for FM [13].

A crucial aspect for including interventions such as those mentioned above in any health system is the balance between costs and benefits that each intervention produces. Policy-makers are faced with limited economic resources and therefore must prioritize among available alternatives. Cost-effectiveness analyses allow cost comparisons of different treatments in relation to the health improvement that each one produces [14]. The cost–utility analysis is a type of cost-effectiveness analysis that allows comparison of therapies across different pathologies, as it is based on quality-adjusted life years (QALYs), a measure that combines the length of survival and its quality, regardless of the illness. The value of a QALY ranges from 0 (death) to 1 (best imaginable health), although states deemed worse than death can have negative values. Thus, cost–utility is measured in costs per QALYs.

Unfortunately, to date, there have been few cost-effectiveness or cost–utility studies of effective interventions for FM, although interest in these has recently increased. To our knowledge, 12 studies have assessed the cost–utility of different treatments for FM [14,15,16,17,18,19,20,21,22,23,24,25]. Four of them focused on pharmacological treatments, concluding that both pregabalin [15,18,19] and duloxetine [20] would be cost-effective compared to other pharmacological options or placebo. Regarding non-pharmacological interventions, one study evaluated the cost–utility of a cognitive-educational treatment for FM and found that the group discussion component alone was more cost-effective as compared to adding a cognitive component [21]. However, further studies have found cognitive-behavioral interventions to be cost-effective. This includes Schröder et al. [25], who studied the long-term cost–utility of a CBT group compared to usual care in functional somatic syndromes such as FM and found that the intervention improved quality of life and reduced costs in the long term.

In another study, Luciano et al. [22] found that the multicomponent intervention FibroQoL, consisting of the combination of psychoeducation and training in relaxation, was cost-effective in the long-term compared to usual care. The same research group found that CBT was more cost-effective than recommended pharmacological treatment (RPT) and usual care due to significant reduction in direct costs, although it was not associated with significantly improved quality of life [23]. More recently, the same authors compared group ACT (GACT) to RPT and a waitlist control (WLC), and the cost–utility analysis favored GACT in comparison to RPT and WLC [14]. Even more recently, Hedman-Lagerlöf et al. [24] evaluated the cost-effectiveness of an internet-based exposure intervention for FM (to stimuli that elicit pain-related distress) compared to WLC and found that the intervention could be highly cost-effective, as each incremental responder generated an annual societal cost reduction of more than $15,000.

Certainly, some treatments show limited cost-effectiveness. For example, in studies of alternative treatments including aquatic training [17] and spa treatments [16], the first concluded that an eight month aquatic training was cost-effective compared to usual care, although some external variables (e.g., distance from the patients’ homes or number of patients that participate in each session) could have a major impact on the cost of the intervention. The second found that the spa treatment improved the quality of life only temporarily but not in the long term and resulted in neither a significant decrease of health care consumption nor in productivity loss.

Regarding MBIs, economic evaluations are scarce and have been specially focused on depression and emotional unstable personality disorder [26] but not on FM. Economic evaluation of MBIs for FM seems particularly relevant because these interventions have demonstrated promising clinical results in previous studies [27,28,29,30], albeit with some methodological limitations, such as small sample sizes or lack of long-term follow-up assessments.

As a response to limitations of previous studies, the EUDAIMON study conducted in Spain recruited a large sample of patients with FM (*N* = 225) and employed a 12 month follow-up evaluation. Results showed superior efficacy of Mindfulness-Based Stress Reduction (MBSR) compared to the multicomponent intervention FibroQoL and to usual care [31]. MBSR led to improved functional impairment, FM-related symptoms, and other secondary outcomes (e.g., depressive and anxiety symptoms, perceived stress, pain catastrophizing, cognitive dysfunction) with moderate to large effect sizes at post-treatment assessment. Some of these improvements were partially lost in the long-term, probably due to reduced, intermittent, and non-structured practice of mindfulness once the eight week intervention finished.

The present study extends our earlier work on clinical efficacy of MBSR in patients with FM [31] and shows the results of an economic evaluation alongside the randomized controlled trial (RCT). Here, we compare for the first time the 12 month health care and societal costs as well as the 12 month cost–utility of MBSR compared to FibroQoL and usual care (passive control arm) in terms of QALY gains and increases in health-related quality of life in Spanish patients with FM.

## 2. Methods

### 2.1. Participants

Following a multi-stage recruitment process, a total of 225 adult patients diagnosed with FM according to the American College of Rheumatology (ACR) 1990 criteria [32] were recruited from the Rheumatology Service at Sant Joan de Déu Hospital (St. Boi de Llobregat, Spain) and participated in the EUDAIMON study between January 2016 and April 2018. As seen in Figure 1, the total sample was randomized into three study arms, as explained below. However, due to missing data in some of the baseline variables needed for the cost–utility analyses (i.e., EQ-5D VAS and/or FM-related medication costs), 21 patients were not included in the economic evaluation, and the final sample of this study consisted of 204 individuals with FM (68 per study arm).

### 2.2. Inclusion and Exclusion Criteria

All 567 potential participants underwent a phone screening to assess the following inclusion criteria: (1) aged between 18–65 years old; (2) able to understand Spanish language; and (3) provided informed consent to participate, and exclusion criteria: (1) participation in a concurrent treatment trial; (2) presence of cognitive impairment according to the Mini Mental State Examination (score < 27) [33]; (3) participation in psychological treatment during the last 12 months; (4) previous experience in meditation or other mind-body therapies; (5) presence of comorbid severe mental or medical disorders that could interfere with treatment; (6) pregnancy; and (7) involvement in ongoing litigation relating to FM.

Those participants meeting the eligibility criteria were scheduled for a first face-to-face interview in the hospital with a trained clinical psychologist blind to treatment allocation where inclusion criteria were checked again; if the criteria were fulfilled, the baseline evaluation started. The evaluations consisted of the administration of a battery of measures to assess different clinical outcomes (e.g., functional impairment, “fibromyalginess”, anxiety and depression symptoms, pain catastrophizing, cognitive dysfunction, and perceived stress), process variables (e.g., mindfulness facets, self-compassion, and psychological inflexibility) and quality of life and cost-related outcomes (e.g., use of clinical services, medication, sick leaves, etc.). Moreover, during the baseline clinical interview, the Structured Clinical Interview for DSM-IV-Axis I Depressive disorders (SCID-I) was used to establish the diagnostic of a current episode of major depression, a previous episode of major depression, and/or dysthymia. See Pérez-Aranda et al. [31] for more detailed information.

### 2.3. Design

The study is registered at Clinicaltrials.gov under registration number NCT02561416. A 12 month RCT was performed with random allocation of the participants to 3 arms (using a computer-generated randomization list): MBSR added to treatment-as-usual (TAU); FibroQoL added to TAU; and TAU alone. A detailed description of the study protocol can be found elsewhere [34]. In summary, all recruited patients signed an informed consent and participated voluntarily in the RCT. This included three assessments: at baseline, post-treatment (or 2 months after baseline, in the case of the participants allocated in the TAU condition), and at 12 months follow-up (48 weeks after randomization).

The study was approved by the Ethics Committee at the Sant Joan de Déu Foundation (PIC-102-15) and was performed in accordance with the ethical standards laid down in the 1964 Declaration of Helsinki and its following updates.

### 2.4. Interventions

#### 2.4.1. MBSR

MBSR is a transdiagnostic program originally developed by Jon Kabat-Zinn [35] to help patients with chronic conditions. Mindfulness is defined as “the awareness that emerges through paying attention on purpose, in the present moment, and nonjudgmentally to the unfolding of experience” [36]. In MBSR, structured training in mindfulness is provided to help patients to relate to their physical and psychological conditions in more accepting and non-judgmental ways [37]. We used the MBSR protocol developed at the University of Massachusetts Medical School (USA) with minimal adaptations for our patients with FM attending to the characteristics of this population. The program consisted of eight weekly 2 h sessions and included the usual one half-day of silent retreat (6 h long session between weeks 6 and 7), although it was optional in our study. The book *Con rumbo propio* [38] and audiotapes were provided to facilitate practice at home, which is reinforced throughout the program and recorded in a practice log. The intervention was delivered in a group format (approximately 15 patients per group), and each group was conducted by a different properly trained MBSR instructor.

#### 2.4.2. FibroQoL

The FibroQoL program is a multicomponent intervention developed by expert and multidisciplinary groups in Catalonia between 2006 and 2007. It was used as an active treatment comparator because it had previously demonstrated cost–utility compared to TAU for FM [22,39]. MBSR and FibroQoL were practically equivalent in terms of structure, which offers a comparison of MBSR to an active control that matches MBSR in non-specific factors but does not contain mindfulness techniques. FibroQoL consists of eight weekly 2 h sessions that are divided in two parts: four sessions of psychoeducation in which patients receive updated information about pathophysiology, diagnosis, and management of FM symptoms, and another four sessions of training in relaxation and self-hypnosis through different techniques with goals to generate a state of deep relaxation, achieve control over the body and pain, and imagine one’s life in the future without pain [12]. Audiotapes were provided to facilitate practice at home. The recently published Beginner’s Guide to Fibromyalgia [40] was also provided for giving updated information about FM syndrome. The intervention was delivered in a group format (15 patients per group), and one team formed by two psychologists, three family physicians, and a rheumatologist conducted the five groups.

#### 2.4.3. TAU

Patients randomized to this arm received no additional active treatment over the study period but continued with their regular pattern of medication (if any). The usual treatment of FM typically includes analgesics, anxiolytics, opioids, antidepressants, and/or anti-inflammatories, and recommendations for practicing aerobic exercise regularly. For ethical reasons, participants allocated to TAU arm were offered participation in an MBI at the end of the study. 

### 2.5. Outcome Measures

#### 2.5.1. The EuroQol Questionnaire [41]

The five-level version of the EuroQol five-dimensional classification system (EQ-5D-5L) is a widely used health-related quality of life instrument with a non-disease-specific classification system that consists of two parts.

In the first part, the patient chooses one of five levels of severity (1 = no problems, 2 = mild problems, 3 = moderate problems, 4 = severe problems, 5 = extreme problems) in five domains: mobility, self-care, usual activities, pain, and anxiety/depression. The time frame is the day of reporting. The combination of the answers given to these domains results in 3125 (5^5^) different health states. The utility scores are obtained from the EQ-5D classification system and are used to rate patients’ health-related quality of life. This continuous variable includes negative values, which indicate a health state “worse than death”, 0, which indicates a state “as bad as death”, and 1, which represents “perfect health”. This scale reflects the health status as described by the subject and is often the preferred method for economic evaluations from a general perspective. In order to derive the EQ-5D utility value from a set of EQ-5D-5L domains, there exist different sets of country-specific preference weights; in our case, these utility values were calculated using the Spanish tariffs of EQ-5D-5L [42], since they are more relevant to our decision-making context. EQ-5D utility values are then used to estimate QALYs, which represent a common measure to assess the outcomes associated with different treatments both in terms of patients’ quality of life and survival [43]. In terms of QALYs, a year of perfect health is worth 1, and a year of less than perfect health is worth less than 1.

The second part of the questionnaire is a visual analogue scale (EQ VAS) on which participants record their current overall health status ranging from 0 (worst imaginable health) to 100 (best imaginable health).

#### 2.5.2. The Client Service Receipt Inventory (CSRI) Spanish Version 

The Client Service Receipt Inventory (CSRI) [44] was used to collect retrospective data on medication and service receipt. For medication intake, patients were asked to bring their daily medication prescriptions and the following information for FM-related drugs (i.e., analgesics, anti-inflammatories, opioids, antiepileptics, muscle relaxants, antidepressants, and anxiolytics) was recorded: the name of the drug, the dosage, the total number of prescription days, and the daily dosage consumed. Regarding service receipt, patients were asked about the total visits to emergency services, the total days of general inpatient hospital admissions, the number of diagnostic tests administered, and the total visits to general practitioner, nurse, social worker, psychologist, psychiatrist, group psychotherapy, and other community health care professionals, specifying in each case if these services were provided by the public or by the private sector. The CSRI was administered on two occasions: at baseline and at 12 month follow-up, both referring to the previous 12 months.

### 2.6. Statistical Analysis

The present economic evaluation is reported according to the Consolidated Health Economic Evaluation Reporting Standards statement [45] (see Appendix A) and follows the Good Research Practices for Cost-Effectiveness Analysis Alongside Clinical Trials [46]. Statistical analyses were performed using STATA v13.0 (StataCorp, College Station, TX, USA) and SPSS v22 (SPSS Inc., Chicago, IL, USA).

We estimated costs from the healthcare and the government perspectives, taking the previous year as the time frame. Catalonia has full governance of health and social care and, as in every other Spanish region, healthcare is universal and publicly financed. The government perspective included direct healthcare costs assumed by the Catalan government (out-of-pocket costs and costs associated with private insurances were not included) and indirect costs related to productivity losses assumed by the Spanish government. The healthcare perspective included only direct healthcare costs. These costs were calculated by summing costs from medications, use of healthcare services, medical tests, and costs of the professionals delivering the MBSR and FibroQoL treatments. We calculated the cost of medications by consulting the price per milligram in the Vademecum International (Red Book; edition 2016) and included the value-added tax. Thus, we computed total costs of medications by multiplying the price per milligram by the total daily dose consumed (in milligrams) and the number of days that the pharmacological treatment was delivered. The SOIKOS database of health care costs [47] was the principal source of unit cost data for health services use and medical tests. The total cost of the MBSR and the FibroQoL treatments took into account the price per patient per group session for the health professional who provided the sessions. Attendance to MBSR and FibroQoL sessions was obtained by consulting professionals’ records. The cost of treatment sessions and resources was considered equal across all sessions and groups, but the number of participants attending those sessions was not; therefore, MBSR and FibroQoL costs were dependent on the number of sessions attended by participants.

We calculated indirect costs (lost productivity) from the human capital approach. We multiplied the minimum daily wage in Spain for 2016 by the number of days on sick leave reported by each patient. Finally, we calculated total costs by summing the direct and the indirect costs. Unit costs are reported in Euros (€) based on 2016 prices. Table 1 displays the unit costs for the calculation of direct and indirect costs. Given that the time horizon was the previous year, it was not necessary to apply a discount to the costs.

The comparison between two intervention groups in the frame of an economic evaluation results in four potential scenarios: (1) the intervention costs less and is more effective than the alternative; (2) the intervention costs more and is less effective than the alternative; (3) the intervention costs less but is less effective than the alternative; and (4) the intervention costs more but is more effective than the alternative. The first two scenarios exhibit strong dominance; thus, the decision on which intervention to adopt is normally straightforward. For the other two scenarios, the decision depends on the incremental cost-effectiveness ratio (ICER), which is defined as the ratio between incremental costs and incremental effects measured on QALYs or EQ VAS points [48]. For considering the intervention cost-effective, each country establishes an investment ceiling, which in the case of Spain is €25,000 per QALY [14].

Our cost-effectiveness analyses were implemented using the Zellner’s seemingly unrelated regression (SUR) model [49]. Estimates were performed using STATA’s sureg command. Using the SUR method for cost-effectiveness purposes implies the use of a bivariate system of regressions that includes both costs and outcomes (with the latter being either QALYs or EQ VAS, depending on the model considered) as the dependent variables of the two separate equations, which are estimated jointly. The regressions of costs and outcomes are therefore part of two regressions on treatment allocation (i.e., whether they were assigned to MBSR, FibroQoL, or TAU) plus an additional set of control variables (measured at baseline): age, gender, marital status, education level, employment status, current episode of major depression, baseline costs, or baseline outcome, depending on the equation considered. Estimates of incremental cost and of incremental effect values using the SUR method described above were derived with 1000 bootstrap replications in order to address a possible skewness in the distribution of the dependent variables [50].

We assessed cost-effectiveness of the interventions using several different scenarios. In the first instance, we performed a complete case analysis (CCA), including only the 128 patients who were assessed both at baseline and at 12 month follow-up. Additional scenarios (sensitivity analysis) adopted instead intention-to-treat (ITT) and per protocol analysis (PPA) approaches. In order to be able to perform an ITT analysis, we needed to impute missing values for those variables that were missing at the 12 month follow-up. In order to do so, we used multiple imputation methods with the chained equations approach [51]. Variables that presented most missing values were, in particular, the EQ-5D-5L domains and the costs of the non-responders at 12 months follow-up. The imputation model, run on ten imputed datasets, included all the main sociodemographic and prognostic variables associated with the outcome variables and the other variables containing missing values. In the present study, patients who had baseline CSRI data (*n* = 204) comprised the ITT sample—missing baseline data were not imputed. Finally, the PPA scenario (2nd sensitivity analysis) was estimated on a sample that included only those who attended at least 6 treatment sessions out of 8, with a final sample size of 107 patients.

## 3. Results

In terms of descriptive statistics, no significant differences were observed between the three study arms in any outcome but the clinical diagnosis of “Current episode of major depression” and “Previous episode of major depression” based on the SCID-I (see Table 2), indicating that the MBSR group had fewer participants currently depressed compared to the other two groups. Considering that this variable could impact the economic evaluation, subsequent analyses were adjusted for “Current episode of major depression” among other covariates. Table 2 displays descriptive details for the sociodemographic variables of this sample.

Table 3 contains the descriptive statistics of costs and outcomes at baseline and at 12 month follow-up, split according to the three arms of the RCT, along with adjusted and unadjusted *p* values.

### 3.1. Baseline Costs

The analyses revealed that only the specialized health care services cost was significantly different among study arms (adjusted *p* value = 0.02), indicating that TAU was the most expensive group in this particular service with an average cost of approximately €660, higher than MBSR (approximately €540) and FibroQoL (approximately €400). However, the other costs did not show any significant difference, including direct and total costs.

### 3.2. Follow-Up Costs

For 12 month follow-up costs, we observed that primary health care services cost was significantly lower for the MBSR group (approximately €200) than for the FibroQoL (approximately €320) and for the TAU groups (approximately €360). Post hoc pairwise comparisons indicated that MBSR’s primary health care costs were significantly lower than TAU’s (adjusted *p* value = 0.002) and presented a marginal significance compared to the FibroQoL group (adjusted *p* = 0.06).

Another marginal significance appeared in the post hoc pairwise comparisons for the variable “Medical tests costs”, indicating that the MBSR group had a lower value than the TAU group (adjusted *p* value = 0.09). As could be expected, the intervention’s cost was also significantly different between the groups, which could be attributed to one group (TAU) not receiving any intervention at all.

The comparisons regarding direct costs did not present statistical significance (adjusted *p* value = 0.13), although post hoc analyses revealed that the MBSR group, with cost at approximately €1160, was significantly lower than the TAU group, with cost at approximately €1600 (adjusted *p* value = 0.02).

Focusing on indirect costs, the analyses revealed no significant differences between the three groups (adjusted *p* value = 0.10), although the post hoc pairwise comparisons indicated that MBSR’s associated indirect costs (approximately €400) were significantly lower than FibroQoL’s (approximately €710, adjusted *p* value = 0.05) and TAU’s (approximately €930, adjusted *p* value = 0.05).

Finally, the total costs were significantly different between the three study arms (adjusted *p* value = 0.04), as the MBSR group was less costly (approximately €1560) compared to the FibroQoL (approximately €2020) and the TAU groups (approximately €2530). Post hoc pairwise analyses showed that the MBSR group had significantly lower total costs compared to the FibroQoL (adjusted *p* value = 0.02) and the TAU groups (adjusted *p* value = 0.02).

### 3.3. Baseline Quality of Life Outcomes

Outcomes at baseline were very similar between the three groups, ranging from 0.48 to 0.53 for the EQ-5D utility scores and between 46 and 47 for the EQ VAS. The pairwise tests revealed no significant differences (*p* > 0.05 in every case).

### 3.4. Follow-Up Quality of Life Outcomes

At 12 month follow-up, the between group differences were significant overall for the EQ-5D utility score (adjusted *p* value = 0.05). The MBSR group had the highest value (0.57), and the TAU group had the lowest value (0.45). On the other hand, no significant differences were observed in the case of the EQ VAS (adjusted *p* value = 0.26). We calculated QALYs based on the EQ-5D utility score, and we found significant differences between the three groups (adjusted *p* value = 0.05).

### 3.5. Cost Utility Analysis from the Government Perspective

As shown in Table 4, MBSR was found to be dominant compared with TAU. The incremental costs (in €) were found to be significant in the base case analysis (completers) and the per-protocol analysis, ranging from approximately €−1030 to €−1110. On the other hand, the incremental effect was significant in the ITT analysis, in which 0.053 QALYs were gained with MBSR compared to TAU. The EQ VAS score, however, did not improve significantly in any case, despite ranging between 7 and 12.

When comparing the two active groups, MBSR showed a significantly lower incremental cost compared to FibroQoL using the completer sample (between €−70 and €−820), but no significant differences were observed in the incremental effects. Here, FibroQoL achieved a slightly better outcome in QALYs that translated into an ICER of €385,400/QALY, which cannot be considered a cost-effective result. Under the other two analyses, the incremental costs did not present any significant difference, ranging from €−540 to €−650. Regarding the incremental effects, they favored MBSR in all the cases, but neither the EQ-5D utility score nor the EQ VAS, which ranged from 7 to 12 depending on the sample used, showed any significant difference.

Finally, the comparison between FibroQoL and TAU indicated that the average incremental cost ranged between €−250 and €−460, with FibroQoL showing lower costs than TAU, although such difference was not found to be significant in any of the three samples considered. On the other hand, the incremental effect for QALYs was significant (0.056) using the completer sample, although the EQ VAS effect was slightly better for the TAU group, resulting in an ICER of €159/EQ VAS points gained, despite not being a significant difference. Under the two other analyses, the incremental effect was also non-statistically significant but favored the FibroQoL group.

Figure 2 shows the degree of uncertainty around the differences in costs and QALYs between the groups from the government perspective in the completer sample.

### 3.6. Cost Utility Analysis from the Health Care Perspective

As shown in Table 5, results were in line with those found when considering the government perspective, whereas incremental costs varied given the different cost aggregated used for this part of the analysis.

When comparing MBSR and TAU, the first was again dominant as the incremental costs were significantly lower, ranging between approximately €−420 and €−490. The incremental effect observed for the QALYs was significant using the ITT sample (0.053, *p* = 0.03).

Incremental costs of MBSR compared to FibroQoL ranged between €−120 and €−280 but were not found to be significantly different. Similarly, the incremental effects did not show any significant difference, although the EQ VAS score ranged from 7 to 12, depending on the sample. All the incremental effects favored MBSR but the incremental QALYs using the completer sample, which resulted in an ICER of €116,300/QALY gained and should not be considered a significant result.

Finally, the incremental costs of FibroQoL compared to TAU ranged between €−190 and €−320, but none of them were significant. The incremental effect in QALYs was found to be significant for the completer sample (0.056). All the incremental effects favored FibroQoL but the incremental EQ VAS using the completer sample, which resulted in an ICER of €121/EQ VAS points gained and should not be considered a significant result.

Although both scenarios presented similar results, it can be observed that, under the health care perspective, the MBSR group achieved a significant reduction in incremental costs compared to TAU in the three samples, including the ITT, which was not significant under the government perspective. On the other hand, the significant reduction in incremental costs of MBSR compared to FibroQoL taking the completer sample was lost when considering the health care perspective.

Appendix A shows the degree of uncertainty around the differences in costs and QALYs between the groups from the health care perspective in the completer sample.

## 4. Discussion

The primary aim of the present study was to analyze the cost–utility of MBSR in a sample of Spanish patients with FM, both from the government and the public health care system perspectives. The intervention was compared to an active control group (i.e., the multicomponent intervention FibroQoL) and to usual care, and the economic evaluation was performed in the context of a 12 month RCT.

The results of this study can be summarized as follows. MBSR (added to TAU) compared to TAU alone was associated with lower direct and total costs in people with FM at 12 month follow-up. This significant decrease of costs was mainly due to a reduction in the costs related to primary health care services and indirect costs during the follow-up period for the MBSR group. The incremental effect on quality of life, measured with QALYs, was significant when considering the ITT sample. Both from the health care and the government perspectives, all ICERs were dominant for MBSR independent of the approach (completers, ITT, or PPA) compared to TAU. These results are similar to those observed in previous studies, as other non-pharmacological interventions have been described as cost-effective when compared with usual care [14,22,23,24,25]. In line with Beard et al. [20], our findings point in the direction of considering that there is a large proportion of patients with FM who remain insufficiently treated with standard pharmacotherapy and could benefit from coadjuvant interventions such as MBSR.

When the two active groups (i.e., MBSR and FibroQoL, both added to TAU) were compared, the only significant difference was observed in the reduction of total costs under the government perspective in favor of MBSR (completers sample). This difference was based primarily on the reduction of indirect costs, as no significant reduction in direct costs was observed (i.e., health care perspective). Reducing indirect costs has been considered as one of the main target points for interventions addressed to chronic pain management [52]. As stated by Hedman-Lagerlöf et al. [24], it is possible that reduction of indirect costs was not only a consequence of reduced symptoms, but that engaging in work-related activities may in turn lead to improvements in FM symptoms. In addition, indirect costs derived from disability, unemployment, and/or early retirement have been associated with disease severity [53], and it would be interesting to update the rates of absenteeism and disability considering the recently proposed classification by Pérez-Aranda et al. [54], as this system already found significant differences in indirect costs among clusters that were not observed using the classical cut-off-based classification method. Considering that not all patients with FM respond equally to every treatment, including MBSR, as the current RCT proved, studying how effective the different, already validated interventions for FM are for each subtype of patient could be the next step toward the ideal of personalized medicine.

In terms of incremental effects on quality of life, no differences were found between MBSR and FibroQoL, indicating that both interventions achieved a similar effect in the long term. This tendency was already observed in the previous study based on this RCT [31], where MBSR was found as clearly more efficacious than FibroQoL at post-treatment, but only significant improvements in fibromyalginess (measured by the Fibromyalgia Survey Diagnostic Criteria [55]) and pain catastrophizing (measured by the Pain Catastrophising Scale [56]) were observed at 12 month follow-up.

It seems reasonable to believe that effects of MBSR on quality of life might show a similar long-term pattern as other outcomes, as it would be intimately related to some of the core FM symptoms (e.g., functional impairment, anxiety and depression, perceived stress, and perceived cognitive dysfunction). Based on what previous studies have demonstrated [57,58,59,60], this partial loss of effect in the long-term could be attributed to a reduction in the frequency of practice of mindfulness exercises once the intervention is over, which would imply that some FM symptoms and presumably quality of life are particularly practice-dependent. Therefore, finding ways to enhance the frequency and the quality of mindfulness home practice is an issue of great relevance to be studied in the future.

The comparison between FibroQoL and TAU indicated that the first produced a significant incremental effect on quality of life on the completers sample, although no significant reduction in costs was observed in any case beyond the perspective considered. Despite being dominant when compared to TAU, FibroQoL was not as superior as it had been in a previous RCT [22] in which the costs were reduced in a similar degree (approximately €−220), but the incremental quality of life was notably higher (0.12). A possible explanation would be that in the previous RCT, the recruited patients were already visited by the same professionals who conducted the FibroQoL program, which could have enhanced the therapeutic alliance, a relevant factor in any treatment context and particularly in a syndrome such as FM that is often associated with the experience of feeling stigmatized [61,62].

Considering previous findings on the cost–utility of different non-pharmacological interventions for FM, we can observe that some, such as the spa treatment or the aquatic training, achieved a significant incremental effect (0.04 and 0.131, respectively) but also higher incremental costs than usual care, resulting in ICERs which ranged between €8000 and more than €30,000 per QALY gained [16,17]. Other interventions, such as GACT [14], achieved a similar improvement in QALYs (0.05 compared to waiting list) as the one that MBSR achieved in the present study compared to TAU and reduced the costs considerably more (approximately €−1900). Also, GACT was dominant compared to recommended drugs under the health care perspective (approximately €−900). It is notable, however, that the GACT group did not consume any medication during the trial, which undoubtedly reduced associated costs. CBT, for its part, significantly reduced the costs compared to TAU (approximately €−2000) and recommended pharmacologic treatment (approximately €−2300), but no significant incremental effect was observed [23]. The STreSS program, however, also a cognitive-behavioral intervention, did achieve a significant incremental effect (0.035) compared to usual care as well as a significant cost reduction in the long term [25]. Finally, the internet-delivered exposure therapy assessed by Hedman-Largelöf et al. [24] not only achieved significant effects (0.07 QALYs gained) but also a great cost reduction (approximately €−5000) compared to usual care. When comparing the incremental effects achieved by the different interventions, one needs to bear in mind that the present study used the EQ-5D-5L, which has been associated with smaller changes in quality of life than the EQ-5D-3L [63], the version that most of the abovementioned studies used [14,17,22,23,24].

On the other hand, if we look at the cost–utility of MBIs for other medical conditions, the systematic review conducted by Duarte et al. [26] concluded that, despite positive results being found for depression and emotional unstable personality disorder, the small number of studies conducted (only five) and the heterogeneity in the interventions (four studies assessed Mindfulness-Based Cognitive Therapy and one MBSR) limited the generalizability of the findings. In this regard, our study extends the existing evidence of the cost–utility of MBIs in this case and for FM the first time.

### Limitations

Some limitations of this study cannot be overlooked. First, given that the economic evaluation was not the primary objective when the original RCT was designed, an unexpectedly higher number of missing baseline data in economic- and quality of life-related variables emerged, invalidating 21 of the original 225 patients for the current study. Moreover, a considerably low follow-up rate (around 65%) added to more missing cost–utility-related data in the 12 month follow-up assessment and yielded a completer sample of only 128 patients. Even though regression models included bootstrapping with 1000 replications to address skewness within the data, the sample size in each study arm did not allow a robust estimation of costs, and confidence intervals were large in most cases; therefore, the results reported should be interpreted with caution.

Another limitation is that the randomization was not stratified by the presence of comorbid major depression, which resulted in the MBSR group having significantly fewer participants with a current episode of major depression compared to the other study arms. However, all reported analyses were performed after adjusting for this variable.

Although it could be thought that public registries would be a better way to collect data on health services use, self-reported data have been demonstrated to be of equal validity as registry-collected data in health-economic assessment [24]. In our study, the CSRI version included recall over a 12 month period, a commonly used time frame in which underreporting is usually more frequent than overreporting due to memory decay and memory biases such as reverse-telescoping (i.e., excluding some events from the recall period) [64]. Some authors, such as Bellón et al. [65], strongly recommend employing recall frames of at least 12 months to reduce memory biases present in patients’ responses in short recall periods. We note that direct non-health care costs including out of pocket expenses, costs of paid and unpaid help, travel expenses, and over the counter medication and other treatment use (e.g., anti-constipation, vitamins, etc.) were not estimated.

Regarding the interventions, it needs to be considered that they were not fully equivalent. MBSR included an optional 6 h retreat, surely increasing the cost of the intervention, which accentuates the significant reduction in total costs that MBSR showed compared to FibroQoL. In terms of program completion, here defined as having attended to at least six of the eight sessions of each program (no retreat included in the case of MBSR), it was low (56% for MBSR and 65% for FibroQoL) but similar to what has been observed in FM intervention studies [28]. This continues to be a difficult problem to solve. Some authors have proposed strategies that could be implemented in further studies, such as written commitments from all participants or makeup classes for those who missed a session [66].

## 5. Conclusions

In summary, the results of the present work support that MBSR (added to TAU) is cost-effective compared with the multicomponent intervention FibroQoL (also added to TAU) and TAU alone. This is mainly because of a reduction in the 12 month follow-up incremental costs (€−1024 compared to TAU and €−771 compared to FibroQoL; government perspective, completers sample) produced essentially in primary health care services and indirect costs. Also, MBSR showed a significant incremental effect in quality of life compared to TAU using the ITT sample (ΔQALYs = 0.053).

FM is a prevalent condition all around the world, however, our results are not necessarily generalizable to all FM patients (our sample has a very small representation of men) nor to other contexts—not only due to cultural differences but also importantly due to differences in how health care systems are organized in other countries. These results support the cost–utility of MBSR for FM, which is in line with previous findings regarding other non-pharmacological interventions such as forms of CBT and ACT. These interventions may have potential to be cost-effective not only for FM but also for treating other chronic pain conditions and/or central sensitivity syndromes (e.g., irritable bowel syndrome, chronic fatigue syndrome, and multiple chemical sensitivity), but this would need specific examination in future studies.

These findings add a substantial contribution to previous studies by presenting, for the first time, an economic evaluation of an MBI for FM. Nonetheless, they should be considered with caution as, among other limitations, the sample of each study arm did not allow robust estimations; if these results were supported by further studies, offering MBSR as a coadjuvant intervention to usual care should be considered as a therapeutic option in the public provision of healthcare.

## Figures and Tables

**Figure 1 jcm-08-01068-f001:**
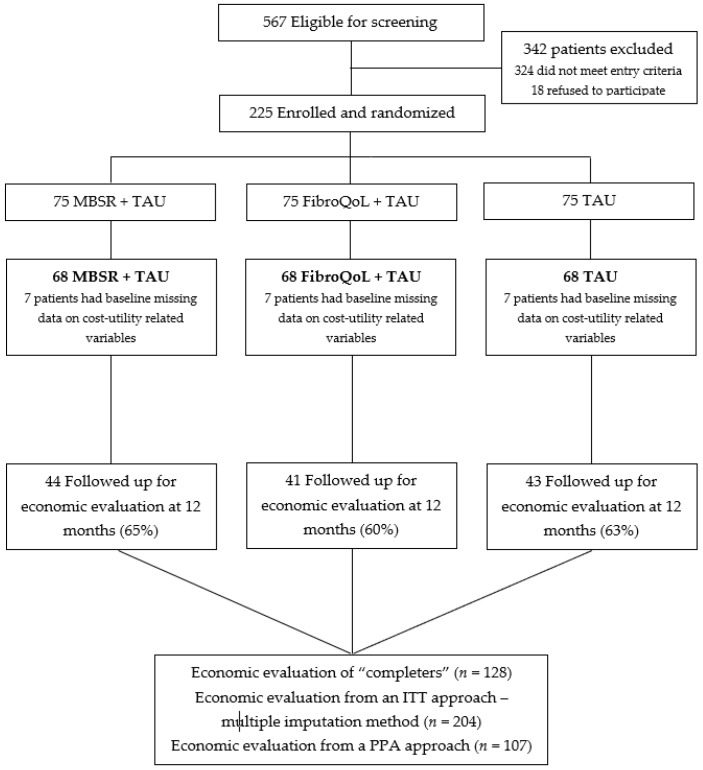
Trial flow chart describing the recruitment process of all three study arms. Note: MBSR = Mindfulness-Based Stress Reduction; TAU = Treatment-as-usual; ITT = intention-to-treat; PPA = Per protocol analysis.

**Figure 2 jcm-08-01068-f002:**
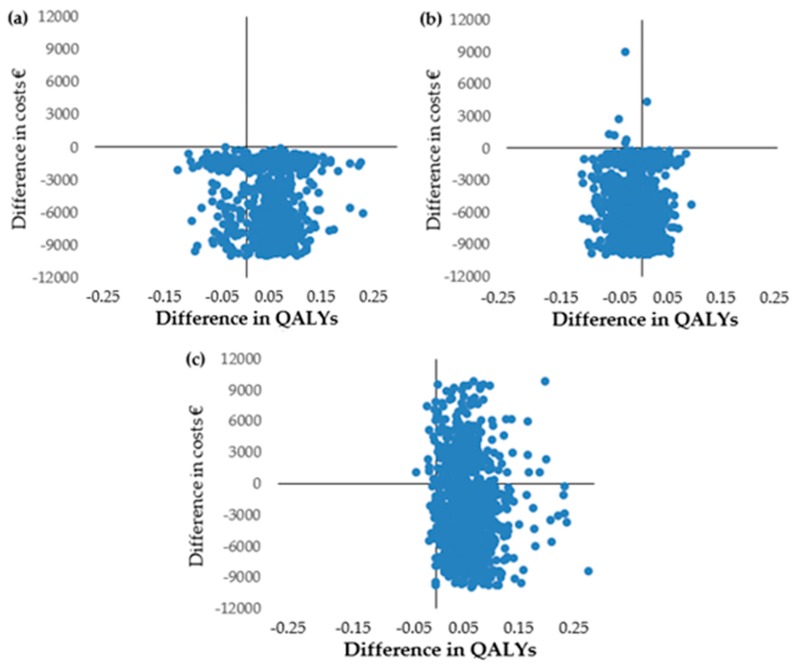
Cost–utility plane of 1000 bootstrap replicated incremental cost–utility from the government perspective (completer sample): (**a**) MBSR vs. TAU; (**b**) MBSR vs. FibroQoL; (**c**) FibroQoL vs. TAU.

**Table 1 jcm-08-01068-t001:** Unit costs used in the calculations of direct and indirect costs (Financial Year 2016; values in €).

Service (Unit)	Costs (€)
Health care General practitioner (per appointment)	
	36.97
Nurse/psychiatric nurse (per appointment)	34.13
Social worker (per appointment)	35.78
Clinical Psychologist (per appointment)	45.06
Psychiatrist (per appointment)	45.06
Other medical specialists (per appointment)	43.82
Accident & Emergency in hospital (per attendance)	99.34
Hospital stay (per night)	112
Diagnostic tests (range)	6.13–455.53
Pharmacological treatment (per daily dose) *	Various
MBSR & FibroQoL (per participant per group session)	45.06
Productivity loss Absenteeism from work (minimum daily wage)	
	21.8

Note: Unit costs were applied to each resource use to compute the total cost of resources used by each participant. All unit costs were for the year 2016. * The cost of prescribed medications was calculated by determining the price per milligram according to the Vademecum International (Red Book; edition 2016) and included the value-added tax.

**Table 2 jcm-08-01068-t002:** Baseline characteristics of patients with Fibromyalgia (FM) by treatment group.

	MBSR (*n* = 68)	FibroQoL (*n* = 68)	TAU (*n* = 68)	*p*
**Gender (women, %)**	66 (97.1%)	67 (98.5%)	67 (98.5%)	0.78
**Age, mean (SD)**	52.63 (8.03)	54.44 (7.69)	53.16 (8.39)	0.4
**Marital status, *n* (%)**				0.59
**Single**	2 (2.9%)	3 (4.4%)	1 (1.5%)	
**Married/living with a partner**	53 (77.9%)	50 (73.5%)	55 (80.9%)	
**Separated/divorced**	11 (16.2%)	9 (13.2%)	10 (14.7%)	
**Widowed**	2 (2.9%)	6 (8.8%)	2 (2.9%)	
**Education level, *n* (%)**				0.21
**Illiterate**	0 (0%)	1 (1.5%)	1 (1.5%)	
**Did not graduate from primary school**	4 (5.9%)	1 (1.5%)	4 (5.9%)	
**Primary school**	31 (45.6%)	37 (54.4%)	32 (47.1%)	
**Secondary school**	31 (45.6%)	24 (35.3%)	28 (41.2%)	
**University**	0 (0%)	5 (7.4%)	3 (4.4%)	
**Others**	2 (2.9%)	0 (0%)	0 (0%)	
**Employment status, *n* (%)**				0.47
**Homemaker**	10 (14.7%)	10 (14.7%)	4 (5.9%)	
**Paid employment**	19 (27.9%)	21 (30.9%)	19 (27.9%)	
**Paid employment but in sick leave**	6 (8.8%)	4 (5.9%)	4 (5.9%)	
**Unemployed with subsidy**	8 (11.8%)	9 (13.2%)	11 (16.2%)	
**Unemployed without subsidy**	8 (11.8%)	15 (22.1%)	9 (13.2%)	
**Retired/pensioner**	9 (13.2%)	4 (5.9%)	10 (14.7%)	
**Temporal disability**	1 (1.5%)	2 (2.9%)	1 (1.5%)	
**Others**	7 (10.3%)	3 (4.4%)	10 (14.7%)	
**Clinical variables**				
**Years of diagnosis, mean (SD)**	14.46 (9.17)	11.28 (7.17)	13.68 (10.02)	0.14
**Current episode of depression, *n* (%)**	24 (35.3%)	39 (57.4%)	38 (55.9%)	0.02
**Previous episode(s) of depression, *n* (%)**	25 (36.8%)	20 (29.4%)	34 (50%)	0.04
**Dysthymia, *n* (%)**	14 (20.6%)	9 (13.2%)	8 (11.8%)	0.31
**Daily FM-related medication**				
**Analgesics, *n* (%)**	21 (30.9%)	21 (30.9%)	15 (22.1%)	0.42
**Anti-inflammatory, *n* (%)**	17 (25%)	17 (25%)	24 (35.3%)	0.31
**Opioids, *n* (%)**	25 (36.8%)	21 (30.9%)	17 (25%)	0.33
**Antiepileptic, *n* (%)**	13 (19.1%)	11 (16.2%)	14 (20.6%)	0.8
**Muscle relaxant, *n* (%)**	2 (2.9%)	5 (7.4%)	3 (4.4%)	0.48
**Antidepressants, *n* (%)**	35 (51.5%)	30 (44.1%)	26 (38.2%)	0.3
**Anxiolytics, *n* (%)**	30 (44.1%)	33 (48.5%)	31 (45.6%)	0.87

**Table 3 jcm-08-01068-t003:** Summary statistics of the costs (total and disaggregated in components) and outcomes according to treatment group.

	MBSR	FibroQoL	TAU	*OMNIBUS* Significance Test	
***Baseline (N = 204)***	*n* = 68	*n* = 68	*n* = 68	*p*	Adjusted *p*
**Primary health care services**	349.6 (325.8)	322.4 (282.8)	316.4 (269.6)	0.78	0.81
**Specialized health care services**	537.6 (438.1)	398.5 (411.9)	661.9 (674.8)	0.01	0.02
**Medical tests**	455.8 (462.8)	474.2 (634.7)	424.3 (480.4)	0.86	0.94
**FM-related medications**	307.6 (488.6)	204.3 (262.6)	171.6 (282.5)	0.15	0.19
**Direct costs**	1650.7 (1069.9)	1399.4 (1006.5)	1574.3 (1220.6)	0.35	0.39
**Indirect costs**	667.8 (1951.1)	669.7 (1569.6)	1144.8 (2953.1)	0.45	0.24
**Total costs**	2318.4 (2417.6)	2069.1 (2075.5)	2719.1 (3783.9)	0.45	0.18
**Outcomes**					
**EQ-5D utility score (0 to 1)**	0.50 (0.21)	0.48 (0.23)	0.53 (0.23)	0.44	0.28
**EQ VAS (0 to 100) ***	46.61 (21.82)	47.31 (19.91)	47.32 (18.51)	0.97	0.75
***12-months Follow-up (N = 128)***	*n* = 44	*n* = 41	*n* = 43		
**Primary health care services**	197.3 (233.4)	319.4 (312.3)	357.9 (301.8)	0.01	0.02
**Specialized health care services**	498 (485.3)	534.1 (552)	664.8 (754.3)	0.45	0.5
**Medical tests**	225.4 (360.3)	257.3 (280.4)	328.2 (417.6)	0.44	0.42
**FM-related medications**	235.9 (349.2)	189.8 (205.1)	255.4 (434.3)	0.56	0.58
**Intervention (MBSR/FibroQoL)**	702.5 (298.9)	578.1 (181.3)	0 (0)	0	0
**Direct costs**	1156.6 (938.3)	1300.6 (872.4)	1598.7 (1265.1)	0.17	0.13
**Indirect costs**	400.9 (1325.2)	714.6 (1905.8)	929.9 (2229.8)	0.36	0.1
**Total costs**	1557.5 (1626.9)	2015.2 (2122.1)	2528.7 (3017)	0.14	0.04
**Outcomes**					
**EQ-5D utility score (0 to 1)**	0.57 (0.25)	0.53 (0.27)	0.45 (0.26)	0.11	0.05
**EQ VAS (0 to 100 points)**	52.41 (23.06)	42.44 (21.16)	44.98 (19.85)	0.09	0.21
**QALY (0 to 1, on the basis of EQ-5D utility score)**	0.54 (0.18)	0.50 (0.20)	0.48 (0.22)	0.34	0.05

Note: Data are presented as mean € cost (SD), except where otherwise is stated. Covariates: age, gender, marital status, education level, employment status, duration of the illness since first diagnostic, current episode of major depression baseline costs, and baseline outcomes, depending on the analyses considered. The mean sessions attended per intervention were 5.3 for MBSR (no retreat included) and 5.8 for FibroQoL. Thirty-two participants (42.7%) attended to the optional mindfulness retreat. ******* One missing value was found in this variable. EQ-5D = EuroQol five-dimensional classification; EQ VAS = visual analogue scale; QALY = quality-adjusted life years.

**Table 4 jcm-08-01068-t004:** Incremental cost, effect, and cost-effectiveness ratios from the government perspective.

	Incremental Cost	Incremental Effect	ICER
	Mean	Mean	
	(95% Bootstrap CI)	(95% Bootstrap CI)	
**MBSR vs TAU**			
**Completers (*n* = 128)**			
QALY (EQ-5D)	**−1023.5** (−2024.7 to −270.5)	0.053 (−0.040 to 0.129)	MBSR dominant
EQ VAS (0-100) *	**−1072** (−2048.5 to −273.6)	7.89 (−1.72 to 18.69)	MBSR dominant
**ITT (*n* = 204)**			
QALY (EQ-5D)	−828.1 (−1699.4 to 43.2)	**0.053** (0.004 to 0.101)	MBSR dominant
EQ VAS (0-100) *	−855.2 (−1727.6 to 17.3)	7.13 (−0.52 to 14.79)	MBSR dominant
**PPA (*n* = 107)**			
QALY (EQ-5D)	**−1036.6** (−1894.3 to −178.9)	0.080 (-0.060 to 0.220)	MBSR dominant
EQ VAS (0–100) *	**−1108.6** (−1968.8 to −248.4)	12.23 (−2.33 to 26.78)	MBSR dominant
**MBSR vs FibroQoL**			
**Completers (*n* = 128)**			
QALY (EQ-5D)	**−770.8** (−1401.4 to −172.4)	-0.002 (-0.066 to 0.059)	€385,400/QALY
EQ VAS (0-100) *	**−822.5** (−1529.1 to −195)	9.46 (-0.84 to 20.35)	MBSR dominant
**ITT (*n* = 204)**			
QALY (EQ-5D)	−539.9 (−1214.6 to 134.8)	0.012 (−0.032 to 0.056)	MBSR dominant
EQ VAS (0-100) *	−575 (−1246.1 to 96.1)	6.68 (−1.01 to 14.37)	MBSR dominant
**PPA (*n* = 107)**			
QALY (EQ-5D)	−582.8 (−1269.1 to 103.4)	0.011 (−0.083 to 0.104)	MBSR dominant
EQ VAS (0-100) *	−651.6 (−1328 to 24.8)	12.08 (−3.62 to 27.78)	MBSR dominant
**FibroQoL vs TAU**			
**Completers (*n* = 128)**			
QALY (EQ-5D)	−252.7 (−1176.6 to 536)	**0.056** (0.006 to 0.172)	FibroQoL dominant
EQ VAS (0-100) *	−249.6 (−1164.5 to 654)	−1.57 (−6.71 to 10.44)	€159/EQ VAS
**ITT (*n* = 204)**			
QALY (EQ-5D)	−288.2 (−1307.9 to 731.6)	0.041 (−0.003 to 0.084)	FibroQoL dominant
EQ VAS (0-100) *	−280.1 (−1297 to 736.6)	0.45 (−7.31 to 8.22)	FibroQoL dominant
**PPA (*n* = 107)**			
QALY (EQ-5D)	−453.8 (−1290.3 to 382.8)	0.069 (-0.010 to 0.149)	FibroQoL dominant
EQ VAS (0-100) *	−456.9 (−1301 to 387.1)	0.15 (−9.47 to 9.76)	FibroQoL dominant

Note: Significant values (*p* < 0.05) are shown in **bold**. Covariates: gender, age, marital status, current episode of major depression, educational level, and employment status. *** Analyses using the EQ VAS score as outcome were computed using one patient less in each case, due to missing data on this baseline variable. ICER = incremental cost-effectiveness ratios.

**Table 5 jcm-08-01068-t005:** Incremental cost, effect, and cost-effectiveness ratios from the health care perspective.

	Incremental Cost	Incremental Effect	ICER
	Mean	Mean	
	(95% Bootstrap CI)	(95% Bootstrap CI)	
**MBSR vs. TAU**			
**Completers (*n* = 128)**			
QALY (EQ-5D)	**−420.7** (−883.8 to −34.9)	0.053 (−0.041 to 0.131)	MBSR dominant
EQ VAS (0–100) *	**−464.8** (−884.2 to −63.3)	7.89 (−1.65 to 18.65)	MBSR dominant
**ITT (*n* = 204)**			
QALY (EQ-5D)	**−455.2** (−904.6 to −4.9)	**0.053** (0.004 to 0.102)	MBSR dominant
EQ VAS (0–100) *	**−483** (−929.5 to −36.5)	7.14 (−0.49 to 14.77)	MBSR dominant
**PPA (*n* = 107)**			
QALY (EQ-5D)	−431.5 (−866.7 to 3.7)	0.080 (−0.060 to 0.220)	MBSR dominant
EQ VAS (0–100) *	**−493.6** (−914.9 to −72.2)	12.22 (−2.57 to 27.02)	MBSR dominant
**MBSR vs FibroQoL**			
**Completers (*n* = 128)**			
QALY (EQ-5D)	−232.6 (−572 to 129.6)	−0.002 (−0.067 to 0.059)	116,300 €/QALY
EQ VAS (0–100) *	−275.2 (−629.2 to 96.7)	9.47 (−1.03 to 20.24)	MBSR dominant
**ITT (*n* = 204)**			
QALY (EQ-5D)	−236.2 (−551.8 to 79.5)	0.012 (−0.033 to 0.058)	MBSR dominant
EQ VAS (0–100) *	−265.2 (−573.9 to 43.6)	6.69 (−0.87 to 14.25)	MBSR dominant
**PPA (*n* = 107)**			
QALY (EQ-5D)	−117.6 (−505.4 to 270.1)	0.011 (−0.083 to 0.104)	MBSR dominant
EQ VAS (0–100) *	−174.2 (−551.3 to 202.8)	12.07 (−3.76 to 27.90)	MBSR dominant
**FibroQoL vs TAU**			
**Completers (*n* = 128)**			
QALY (EQ-5D)	−188 (−696.2 to 227.8)	**0.056** (0.007 to 0.173)	FibroQoL dominant 121 €/EQ VAS
EQ VAS (0–100) *	−189.6 (−625.8 to 259.4)	−1.57 (−10.49 to 6.68)	
**ITT (*n* = 204)**			
QALY (EQ-5D)	−219.1 (−656.3 to 218.1)	0.041 (−0.002 to 0.084)	FibroQoL dominant
EQ VAS (0–100) *	−217.8 (−655.9 to 220.3)	0.45 (−7.22 to 8.12)	FibroQoL dominant
**PPA (*n* = 107)**			
QALY (EQ-5D)	−313.9 (−747.9 to 120.2)	0.069 (−0.010 to 0.149)	FibroQoL dominant
EQ VAS (0–100) *	−319.4 (−748.8 to 110.1)	0.15 (−9.42 to 9.73)	FibroQoL dominant

Note: Significant values (*p* < 0.05) are shown in **bold**. Covariates: sex, age, marital status, current episode of major depression, educational level, and employment status. *** Analyses using the EQ VAS score as outcome were computed using one patient less in each case due to missing data on this baseline variable.

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
