# Peer review of "Cost–Utility of Mindfulness-Based Stress Reduction for Fibromyalgia versus a Multicomponent Intervention and Usual Care: A 12-Month Randomized Controlled Trial (EUDAIMON Study)"

_jcm, 2019, doi:10.3390/jcm8071068_

Reviewer 1 Report

The authors proposed an interesting and relevant paper about the costs and effects of a mindfulness-based stress reduction intervention as compared to a multicomponent intervention or usual care. The following suggestions (minor) may help the authors to improve the clarity of their paper: 

Abstract – “Usual care” vs “TAU”. The acronym should be adjusted or defined.

Abstract – Perspective of the economic analysis should be clearly specified (e.g., public healthcare system, third party payers, patient, societal).

Abstract – The clarity of the sentence “It also produced a significant difference on QALY gains compared to TAU (0.053)”. should be improved. It is not clear for the reader if 0.053 is the QALY gain over time in the MBSR group or the difference between study groups. In the results, effect sizes for the different groups and statistical significance should be presented.  

Introduction, lines 79-80 – It is reported that “cost-utility is measured in QALYs divided by the incremental cost”. The sentence is incorrect since cost-utility is the cost per QALY, or incremental cost-effectiveness ratio, which is calculated as the difference between the costs of the 2 interventions, divided by the difference in QALYs between the two interventions.

Methods, inclusion and exclusion criteria – How was the presence of FM assessed?

Methods, lines 247-248 – It is stated that “the economic evaluation was carried out following the Consolidated Health Economic Evaluation Reporting Standards statement…”. Since that statement presents reporting guidelines, authors should rather say that the paper was written according to these guidelines and provide a completed CHEERS checklist (http://www.equator-network.org/reporting-guidelines/cheers/).

Methods, Client Service Receipt Inventory (CSRI) – Please cover the validity of a 12 months’ time window regarding recall bias (here or in the discussion). See: (Bhandari & Wagner, 2006)

Methods, lines 233-273 – The authors adopted healthcare and government perspective. However, some information is lacking to allow the international reader to understand which costs were considered: Is the healthcare system universal? Does it cover all direct costs for everybody? Are out-of-pocket costs or costs covered by private insurances considered in the CSRI? Where those costs excluded from the analysis since a government perspective was adopted?

Methods, lines 282-294 – This information should be combined to the section presenting the EQ-5D questionnaire.

Results – Please see the following editorial about p-values in baseline tables of randomized controlled trials: (Knol et al., 2012)

The discussion is complete and well written. Main limitations are covered, but external validity of results (other contexts, other FM population should be discussed).

The authors state that “the sample size in each study arm did not allow a robust estimation of costs”. Could this be explained further? What would be the ideal sample size? Was the needed sample size for the cost analysis calculated a priori and considered in the RCT?

References

Bhandari, A., & Wagner, T. (2006). Self-reported utilization of health care services: improving measurement and accuracy. Medical Care Research and Review, 63(2), 217-235.

Knol, M. J., Groenwold, R. H., & Grobbee, D. E. (2012). P-values in baseline tables of randomised controlled trials are inappropriate but still common in high impact journals. Eur J Prev Cardiol, 19(2), 231-232.

Author Response

Thank you for your comments, we believe that your suggestions have helped to improve the quality of our work. We hope that you find all the changes added to the manuscript to be satisfactory.

Abstract – “Usual care” vs “TAU”. The acronym should be adjusted or defined.

Authors: Thank you for noticing it, this has been fixed.

Line 30: “…as add-on to treatment-as-usual (TAU) for patients…”.

Abstract – Perspective of the economic analysis should be clearly specified (e.g., public healthcare system, third party payers, patient, societal).

Authors: This information has been added.

Lines 33-34: “…were included in the cost-utility analysis, which were conducted both under the government and the public healthcare system perspectives.

Abstract – The clarity of the sentence “It also produced a significant difference on QALY gains compared to TAU (0.053)”. should be improved. It is not clear for the reader if 0.053 is the QALY gain over time in the MBSR group or the difference between study groups. In the results, effect sizes for the different groups and statistical significance should be presented.  

Authors: We have rewritten that fragment to clarify that we were referring to the incremental effect when comparing MBSR vs TAU and to add the information regarding the statistical significance:

Lines 37-42: “Two sensitivity analyses (intention-to-treat, ITT, and per-protocol, PPA) were conducted. The results indicated that MBSR achieved a significant reduction in costs compared to the other study arms (p< 0.05 in the completers sample), especially in terms of indirect costs and primary healthcare services. It also produced a significant incremental effect compared to TAU in the ITT sample (ΔQALYs= 0.053, p< 0.05).

Introduction, lines 79-80 – It is reported that “cost-utility is measured in QALYs divided by the incremental cost”. The sentence is incorrect since cost-utility is the cost per QALY, or incremental cost-effectiveness ratio, which is calculated as the difference between the costs of the 2 interventions, divided by the difference in QALYs between the two interventions.

Authors: You are completely right, thank you for noticing it. We have fixed this error.

            Line 82-83: “Thus, cost-utility is measured in costs per QALYs”.

Methods, inclusion and exclusion criteria – How was the presence of FM assessed?

Authors: This information has been added.

Line 135: “…diagnosed with FM according to the American College of Rheumatology (ACR) 1990 criteria [32] were recruited…”

Methods, lines 247-248 – It is stated that “the economic evaluation was carried out following the Consolidated Health Economic Evaluation Reporting Standards statement…”. Since that statement presents reporting guidelines, authors should rather say that the paper was written according to these guidelines and provide a completed CHEERS checklist (http://www.equator-network.org/reporting-guidelines/cheers/).

Authors: Thank you for pointing that out. The sentence has been corrected and the completed CHEERS checklist is now available as supplementary material.

            Line 260: “The present economic evaluation is reported according to the…”

Line 603-604: “www.mdpi.com/xxx/s2, Table S1: Consolidated Health Economic Evaluation Reporting Standards Checklist

Methods, Client Service Receipt Inventory (CSRI) – Please cover the validity of a 12 months’ time window regarding recall bias (here or in the discussion). See: (Bhandari & Wagner, 2006)

Authors: This is a very valuable addition which we have included in the limitations section.

Lines 562-567: “In our study, the CSRI version included recall over a 12-month period, a commonly used time frame in which underreporting is usually more frequent than overreporting due to memory decay and memory biases such as reverse-telescoping (i.e., excluding some events from the recall period) [63]. Some authors, such as Bellón et al, [64] strongly recommend employing recall frames of at least 12 months to reduce memory biases present in patients’ responses in short recall periods.

Methods, lines 233-273 – The authors adopted healthcare and government perspective. However, some information is lacking to allow the international reader to understand which costs were considered: Is the healthcare system universal? Does it cover all direct costs for everybody? Are out-of-pocket costs or costs covered by private insurances considered in the CSRI? Where those costs excluded from the analysis since a government perspective was adopted?

Authors: We agree that this information is relevant, and we have added some clarifications in the Methods section:

Lines 264-269: “We estimated costs from the healthcare and government perspectives taking the previous year as time frame. Catalonia has full governance of health and social care and, as in every other Spanish region, healthcare is universal and publicly financed. The government perspective included direct healthcare costs assumed by the Catalan government (out-of-pocket costs and costs associated to private insurances were not included) and indirect costs related to productivity loses assumed by the Spanish government.”

Methods, lines 282-294 – This information should be combined to the section presenting the EQ-5D questionnaire.

Authors: Following your suggestion, we have moved that paragraph to the 2.5.1 section, lines 227-241.

Results – Please see the following editorial about p-values in baseline tables of randomized controlled trials: (Knol et al., 2012)

Authors: We agree with the reviewer that baseline differences are not necessary to be tested for statistical significance because if present, they are probably due to chance. But, at the same time, we find it important to examine for between-group clinically relevant differences at baseline and, if necessary, they should be adjusted for. In our case, it was especially relevant for the subsequent analyses to detect that there was baseline difference in the presence of “Current episode of major depression” given that this variable could impact the economic analyses in a significant manner.

The discussion is complete and well written. Main limitations are covered, but external validity of results (other contexts, other FM population should be discussed).

Authors: Thank you, we have added the following paragraph to address the topics you suggest:

Lines 586-594: “FM is a prevalent condition all around the world, however, our results are not necessarily generalizable to all FM patients (our sample has a very small representation of men) nor to other contexts – not only due to cultural differences but also, importantly, due to differences in how health care system are organized in other countries. These results support the cost-utility of MBSR for FM, which is in line with previous findings regarding other non-pharmacological interventions such as forms of CBT and ACT. These interventions may have potential to be cost-effective, not only for FM, but also for treating other chronic pain conditions and/or central sensitivity syndromes (e.g., irritable bowel syndrome, chronic fatigue syndrome and multiple chemical sensitivity), but this would need specific examination in future studies.”

The authors state that “the sample size in each study arm did not allow a robust estimation of costs”. Could this be explained further? What would be the ideal sample size? Was the needed sample size for the cost analysis calculated a priori and considered in the RCT?

Authors: The sample size needed was calculated considering the effectiveness analyses, which have been published somewhere else (Pérez-Aranda et al., 2019 in PAIN):

“Following Schmidt et al.’s protocol [74] sample size was established on the basis of a previous meta-analysis of controlled MBSR trials [29] in which a mean effect size of d= 0.53 was found. This effect size results in a statistical power of (1-b) = 0.80 for N= 75 patients per group (225 patients overall) assuming a dropout of 25%.”

Unfortunately, as explained in the Limitations section, in the present study there was a notable loss of cost-related data, primarily regarding the FM-related medication pattern data. For this reason, twenty-one patients were not even included in the baseline evaluation, which implied that the total sample was reduced from 225 to 204. Also, the dropout rates were higher than expected (37%).

In summary, if the study had been conducted using the original sample (225) and the dropout ratio had been adjusted to the 25% that was estimated, the results would have been undoubtedly more robust.

References

29. Grossman P, Niemann L, Schmidt S, Walach H. Mindfulness-based stress reduction and health benefits: A meta-analysis. J Psychosom Res 2004;57:35–43. doi:10.1016/S0022-3999(03)00573-7.

74. Schmidt S, Grossman P, Schwarzer B, Jena S, Naumann J, Walach H. Treating fibromyalgia with mindfulness-based stress reduction: Results from a 3-armed randomized controlled trial. Pain 2011;152:361–369. doi:10.1016/j.pain.2010.10.043.

Reviewer 2 Report

All the sections are detailed, well described and clear. 

I think that the “Intervention” and “outcome measures” sections should be more synthetic. 

The introduction is clear, and the rationale is strongly motivated with a solid background. 

Regarding methodology, methods are well described and the recruitment process table is intuitive. However, abbreviations should be explained in the legend, also. 

It would be interesting to implement other outcome measures in the paper, such as for example the impact of the mindfulness based protocol on patient's quality of life.

Furthermore, the part relating to costs is perhaps too abundant compared to the part concerning the impact of mindfulness on the quality of life of the patient with fibromyalgia.

Regarding discussion section,I suggest to add a few sentences to introduce the topic (a sort of brief introduction), before summarizing the results.

The conclusion section highlights the strengths and the limitations in a clear way; however, I’d expand this paragraph with a focus on significant data emerged.

Author Response

Thank you for your comments, we believe that your suggestions have helped to improve the quality of our work. We hope that you find all the changes added to the revised version of the manuscript to be satisfactory.

All the sections are detailed, well described and clear. 

I think that the “Intervention” and “outcome measures” sections should be more synthetic.

Authors: Thank you. We have shortened the section “Interventions” following your suggestion, leaving the information that we consider more relevant. See lines 180 to 220.

On the other hand, the section “Outcomes” now includes a paragraph that was located in the “Statistical Analysis” section, following Reviewer 1’s suggestion. We truly believe that the information provided is important to follow the analyses and results that are presented in the next sections of the paper.

The introduction is clear, and the rationale is strongly motivated with a solid background. 

Regarding methodology, methods are well described and the recruitment process table is intuitive. However, abbreviations should be explained in the legend, also. 

Authors: Thank you. Following your suggestion, we have added the abbreviations in the legend of Figure 1.

Line 145-146:Note: MBSR= Mindfulness-Based Stress Reduction; TAU= Treatment-as-usual; ITT= intention-to-treat; PPA= Per protocol analysis

It would be interesting to implement other outcome measures in the paper, such as for example the impact of the mindfulness based protocol on patient's quality of life.

Authors: We appreciate your suggestion. The study is focused on quality of life, as we have used the EQ-5D as main outcome. No other measure of quality of life was considered in this RCT, although some other FM-related measures were used. There is a recently accepted for publication paper that reports the effectiveness of MBSR on functional impact, “fibromyalginess”, anxiety and depression, perceived stress, pain catastrophizing and cognitive dysfunction (see Pérez-Aranda et al., 2019 in PAIN). This paper also reports the mediational effect of process variables such as mindfulness facets and psychological inflexibility.

Furthermore, the part relating to costs is perhaps too abundant compared to the part concerning the impact of mindfulness on the quality of life of the patient with fibromyalgia.

Authors: Although we agree with you on the fact that the two sections are not balanced in terms of length, this is due to the fact that the “costs” section includes much more information (primary health, specialists, FM-related drugs, medical tests, intervention, direct, indirect and total costs) than the “outcomes” section, which only refers to utilities and EQ-VAS points. Considering that the cost-related information is relevant, we cannot find a better way to report it, accepting the abovementioned length unbalance.

Regarding discussion section, I suggest to add a few sentences to introduce the topic (a sort of brief introduction), before summarizing the results.

Authors: A brief introduction to the Discussion has been added.

Lines 460-464: “The primary aim of the present study was to analyze the cost-utility of MBSR in a sample of Spanish patients with FM, both from the government and the public health care system perspectives. The intervention was compared to an active control group (i.e., the multicomponent intervention FibroQoL) and to usual care, and the economic evaluation was performed in the context of a 12-month RCT.

The conclusion section highlights the strengths and the limitations in a clear way; however, I’d expand this paragraph with a focus on significant data emerged.

Authors: Thank you, we totally agree with you that the conclusion needed to focus more on the significant results. Following your suggestion, we have added that information:

Lines 580-585: “This is mainly because of a reduction in the 12-month follow-up incremental costs (€-1,024 compared to TAU and €-771 compared to FibroQoL; government perspective, completers sample), produced essentially in primary health care services and indirect costs. Also, MBSR showed a significant incremental effect in quality of life compared to TAU using the ITT sample (ΔQALYs= 0.053)."
